# Token Assorted: Mixing Latent and Text Tokens for Improved Language Model Reasoning

DiJia Su [1]   Hanlin Zhu [*2]   Yingchen Xu [*13]   Jiantao Jiao [2]   Yuandong Tian [†1]   Qinqing Zheng [†1]

## Abstract

Large Language Models (LLMs) excel at reasoning and planning when trained on chain-of-thought (CoT) data, where the step-by-step thought process is explicitly outlined by text tokens. However, this results in lengthy inputs where many words support textual coherence rather than core reasoning information, and processing these inputs consumes substantial computation resources. In this work, we propose a hybrid representation of the reasoning process, where we partially abstract away the initial reasoning steps using latent discrete tokens generated by VQ-VAE, significantly reducing the length of reasoning traces. We explore the use of latent trace abstractions in two scenarios: 1) training the model from scratch for the Keys-Finding Maze problem, 2) fine-tuning LLMs on this hybrid data with an extended vocabulary including unseen latent tokens, for both logical and mathematical reasoning problems. To facilitate effective learning, we introduce a simple training procedure that randomly mixes latent and text tokens, which enables fast adaptation to new latent tokens. Our approach consistently outperforms the baselines methods in various benchmarks, such as Math (+4.2%, Llama-3.2-1B), GSM8K (+4.1%, Llama-3.2-3B), and Fresh-Gaokao-Math-2023 (+13.3%, Llama-3.1-8B) with an average reduction of 17% in reasoning trace's length.

## 1 Introduction

Reasoning capabilities are increasingly recognized as a critical component of Artificial General Intelligence (AGI) systems. Recent research has demonstrated that Large Language Models (LLMs) can exhibit sophisticated reasoning and planning abilities using chain-of-thought (CoT) methodologies, including prompting LLMs with examples where complex problems are broken down into explicit reasoning steps (Wei et al., 2022b; Chen et al., 2022a; Yao et al., 2024). More recently, a number of studies have further shown that when models are trained to articulate the intermediate steps of a reasoning process (Nye et al., 2021b; Lehnert et al., 2024), they achieve significantly higher accuracy. The effectiveness of this approach has been demonstrated across multiple domains, including mathematical problem-solving (Yue et al., 2023; Gandhi et al., 2024; Yu et al., 2023; Tong et al., 2024), logical inference (Lin et al., 2024; Dziri et al., 2024), multistep planning tasks (Lehnert et al., 2024; Su et al., 2024), etc.

However, training with explicit reasoning traces in text space comes with notable computational costs (Deng et al., 2023; 2024), as the models must process lengthy input sequences. In fact, much of the text serves primarily to maintain linguistic coherence, rather than conveying core reasoning information. Several works have attempted to mitigate this issue. For example, Hao et al. (2024) investigate reasoning in continuous latent space as a means of compressing the reasoning trace, and Deng et al. (2024) explore internalizing the intermediate steps through iterative CoT eliminations, see Section 2 for more examples. Nonetheless, these approaches rely on multi-stage training procedures that resemble curriculum learning, which still incur significant computational costs, and their final performances fall behind models trained with complete reasoning traces.

To tackle this challenge, we propose to use discrete latent tokens to abstract the initial steps of the reasoning traces. These latent tokens, obtained through a vector-quantized variational autoencoder (VQ-VAE), provide a compressed representation of the reasoning process by condensing surface-level details. More precisely, we replace the text tokens with their corresponding latent abstractions from left to right until a pre-set location, leaving the remaining tokens unchanged. We then fine-tune LLMs with reasoning traces with such *assorted tokens*, allowing the models to learn from both abstract representations of the thinking process and detailed textual descriptions. One technical

---

*Equal contribution †Equal advising [1]Meta AI [2]UC Berkeley [3]UCL. Correspondence to: DiJia Su, Yuandong Tian, Qinqing Zheng.

*Proceedings of the 42nd International Conference on Machine Learning*, Vancouver, Canada. PMLR 267, 2025. Copyright 2025 by the author(s).

challenge posed for the fine-tuning is that the vocabulary is now extended and contains unseen latent tokens. To facilitate quick adaptation to those new tokens, we employ a *randomized replacement* strategy: during training, we randomly vary the number of text tokens being substituted by latent tokens for each sample. Our experiments confirm that this simple strategy leads to straightforward accommodation of unseen latent tokens.

We conduct a comprehensive evaluation of our approach on a diverse range of benchmarks spanning multiple domains. Specifically, we assess its performance on multi-step planning tasks (Keys-Finding Maze) and logical reasoning benchmarks (ProntoQA (Saparov & He, 2022), ProsQA (Hao et al., 2024)) for training T5 or GPT-2 models from scratch. In addition, we fine-tune different sizes of LLama-3.1 and LLama-3.2 models using our approach and evaluate them on a number of mathematical reasoning benchmarks, including GSM8K (Cobbe et al., 2021a), Math (Hendrycks et al., 2021), and OlympiadBench-Math (He et al., 2024), see Section 4.2 for more details. Across all these tasks and model architectures, our models consistently outperform baseline models trained with text-only reasoning traces, demonstrating the effectiveness of compressing the reasoning process with assorted tokens.

## 2 Related Work

**Explicit Chain-of-Thought Prompting.** The first line of work in Chain-of-Thought (CoT) use the traditional chain of prompt in text tokens (Wei et al., 2022a; Nye et al., 2021a). Research works demonstrated that by adding few-shot examples to the input prompt or even zero-shot, the model can perform better in question answering (Chen et al., 2022b; Kojima et al., 2022; Chung et al., 2024). To further improve the model reasoning performance, there has been research effort into prompting with self-consistency (Wang et al., 2022). Here the model is prompted to generate multiple responses and select the best one based on majority voting. On the other hand, research has shown that top-$k$ alternative tokens in the beginning of the prompt can also improve the model's reasoning capability (Wang & Zhou, 2024). On top of these empirical results, there has been research on theoretical understanding of why CoT improves the model's performance through the lens of expressivity (Feng et al., 2024; Li et al., 2024) or training dynamics (Zhu et al., 2024). In a nutshell, CoT improves the model's effective depth because the generated output is being fed back to the original input. CoT is also important for LLMs to perform multi-hop reasoning according to the analysis of training dynamics (Zhu et al., 2024).

**Learning with CoT Data.** In addition to the success of CoT prompting, an emerging line of works have explored training LLMs on data with high-quality reasoning traces, for example, the works of Nye et al. (2021b); Azerbayev et al. (2023); Lehnert et al. (2024); Su et al. (2024); Yu et al. (2024); Yang et al. (2024); Deng et al. (2023; 2024). There is also a surge of interest in synthesizing datasets with diverse intermediate steps for solving problems in various domains, see, e.g., the works of Kim et al. (2023); Tong et al. (2024); Yu et al. (2023); Yue et al. (2023); Lozhkov et al. (2024). Wen et al. (2024) also theoretically studies how training with reasoning trace can improve the sample complexity of certain tasks.

**LLM Reasoning in Latent Space.** There has been research investigating LLM reasoning in the latent space. Hao et al. (2024) have proposed to use the last hidden state of a language model as the next input embeddings, allowing the model to continue reasoning within a continuous latent space. The authors show that this approach effectively captures multiple reasoning paths simultaneously, mimicking a breadth-first-search strategy. Goyal et al. (2023) proposes to insert learnable pause tokens into the original text, in order to delay the generation. As a result, the model can leverage additional computation before providing the final answer. Parallel to this, Pfau et al. (2024) have explored filler tokens, which are used to solve computational tasks that are otherwise unattainable without intermediate token generation. In addition, Liu et al. (2024) propose a latent coprocessor method that operates on the transformer's key-value cache to improve the LLM performance. Nevertheless, none of these methods have shown good performance when integrated into modern-sized LLMs and tested on real-world LLM datasets instead of synthetic ones. Also, Wang et al. (2023) proposed to use the planning token at the start of generation. Orthogonal to these works, Pagnoni et al. (2024) proposes a tokenization-free architecture that encodes input bytes into continuous patch representations, which is then used to train a latent Transformer, and Barrault et al. (2024) perform autoregressive sentence prediction in an embedding space. While these two works both leverage continuous latent spaces, our work focuses on the direct use of discrete latent tokens.

## 3 Methodology

In this section, we describe our methodology to enable LLMs to reason with discrete latent tokens. The notations are summarized in Appendix B. Let $X = P \oplus C \oplus S$ denote a sample input, where $P = (p_1, p_2, \ldots, p_{t_p})$ are the prompt tokens, $C = (c_1, c_2, \ldots, c_{t_c})$ are the reasoning step (chain-of-thought) tokens, $S = (s_1, s_2, \ldots, s_{t_s})$ are the solution tokens, and $\oplus$ denotes concatenation. Our training procedure consists of two stages:

1. **Learning latent discrete tokens to abstract the rea-**

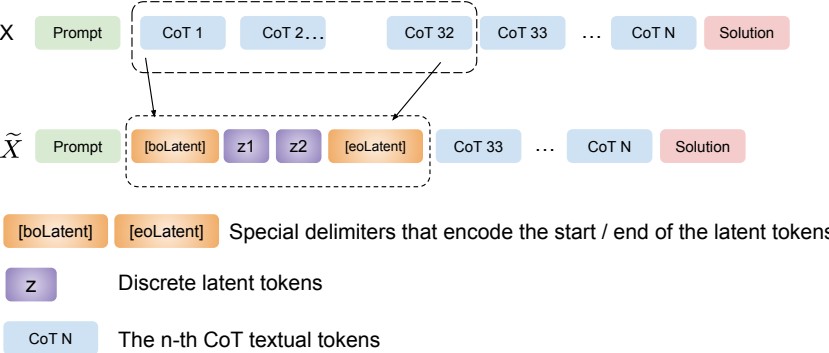

Figure 3.1: An example illustrating our replacement strategy. With chunk size $L = 16$ and compression rate $r = 16$, we encode 32 textual CoT tokens into 2 discrete latent tokens from left to right. The other CoT tokens will remain in their original forms.

soning steps, where we train a model to convert $C$ into a sequence of latent tokens $Z = (z_1, z_2, \ldots, z_{t_z})$ such that $t_z < t_c$. The compression rate $r = t_c/t_z$ controls the level of abstraction.

2. **Training the LLM with a partial and high-level abstract of the reasoning steps**, where we construct a modified input $\widetilde{X}$ by replacing the first $m$ tokens of $C$ by the corresponding latent abstractions:

$$\widetilde{X} = P \oplus [z_1, \ldots, z_{\frac{m}{r}}, c_{m+1}, \ldots, c_{t_c}] \oplus S. \quad (1)$$

Figure 3.1 illustrates this replacement strategy. We randomize the value of $m$ during training.

### 3.1 Learning Latent Abstractions

We employ a vector-quantized variable autoencoder (VQ-VAE) (Van Den Oord et al., 2017) type of architecture to map CoT tokens $C$ into discrete latent tokens $Z$. To enhance abstraction performance, our VQ-VAE is trained on the whole input sequence $X$, but only applied to $C$ in the next stage. Following Jiang et al. (2022; 2023), we split $X$ into chunks of length $L$ and encode each chunk into $\frac{L}{r}$ latent codes, where $r$ is a preset compression rate. More precisely, our architecture consists of the following five components:

- $\mathcal{E}$ : a codebook containing $|\mathcal{E}|$ vectors in $\mathbb{R}^d$.

- $f_{\text{enc}} : \mathcal{V}^L \mapsto \mathbb{R}^{d \times \frac{L}{r}}$ that encodes a sequence of $L$ text tokens to $\frac{L}{r}$ latent embedding vectors $\bar{X} = \bar{x}_1, \ldots, \bar{x}_{\frac{L}{r}}$, where $\mathcal{V}$ is the vocabulary of text tokens.

- $q : \mathbb{R}^d \mapsto \mathcal{E}$: the quantization operator that replaces the encoded embedding $\bar{x}$ by the nearest neighbor in $\mathcal{E}$: $q(\bar{x}) = \text{argmin}_{e_i \in \mathcal{E}} \|e_i - \bar{x}\|_2^2$.

- $g : \mathcal{V}^K \mapsto \mathbb{R}^d$ that maps $K$ text tokens to a $d$-dimensional embedding vector. We use $g$ to generate a continuous embedding of the prompt $P$.

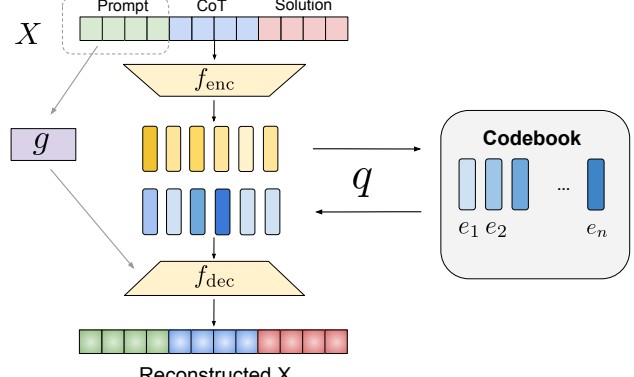

Figure 3.2: A graphical illustration of our VQ-VAE. $f_{\text{enc}}$ encodes the text tokens into latent embeddings, which are quantized by checking the nearest neighbors in the codebook. $f_{\text{dec}}$ decodes those quantized embeddings back to text tokens. When applying the VQ-VAE to compress the text tokens, the discrete latent tokens $Z$ are essentially the index of corresponding embeddings in the codebook.

- $f_{\text{dec}} : \mathbb{R}^{d \times \frac{L}{r}} \times \mathbb{R}^k \mapsto \mathcal{V}^L$ that decodes latent embeddings back to text tokens, conditioned on prompt embedding.

In particular, each continuous vector $e \in \mathcal{E}$ in the codebook has an associated latent token $z$, which we use to construct the latent reasoning steps $Z$[1].

For simplicity, we assume the lengths of the input $X$ and the prompt $P$ are $L$ and $K$ exactly. Similar to Van Den Oord

---

[1]To decode a latent token $z$, we look up the corresponding embedding $e \in \mathcal{E}$ and feed it to $f_{\text{dec}}$.

et al. (2017), we use an objective $\mathcal{L}$ composed of 3 terms:

$$\mathcal{L}(X) = \underbrace{\log p(X|f_{\text{dec}}(q(\bar{X})|g(P)))}_{\text{reconstruction loss}} +$$

$$\sum_{i=1}^{L} \underbrace{\|\text{sg}[\bar{X}_i] - q(\bar{X}_i)\|_2^2}_{\text{VQ loss}} + \underbrace{\beta\|\bar{X}_i - \text{sg}[q(\bar{X}_i)]\|_2^2}_{\text{commitment loss}}, \quad (2)$$

where $\bar{X} = f_{\text{enc}}(X)$, $\text{sg}[\cdot]$ is the stop-gradient operator, and $\beta$ is a hyperparameter controlling the strength of the commitment loss. The VQ loss and the commitment loss ensure that the encoder outputs remain close to the codebook, while the reconstruction loss concerns with the decoding efficacy. As standard for VQ-VAE, we pass the gradient $\nabla_{f_{\text{dec}}}(L)$ unaltered to $f_{\text{enc}}$ directly as the quantization operator $q(\cdot)$ is non-differentiable. Figure 3.2 illustrates our architecture. In practice, we use a causal Transformer for both $f_{\text{enc}}$ and $f_{\text{dec}}$, the model details are discussed in Appendix A.

Thus far we obtain a latent representation both semantically meaningful and conducive to reconstruction, setting the stage for the subsequent training phase where the LLM is trained to perform reasoning with abstractions.

## 3.2 Reasoning with Discrete Latent Tokens

In this second stage, we apply the obtained VQ-VAE to form modifed samples $\widetilde{X}$ with latent abstractions as in Equation (1), then train an LLM to perform next token prediction. Below, we outline the major design choices that are key to our model's performance, and ablate them in Section 4.3.

**Partial Replacement.** Unlike previous planning works (Jiang et al., 2022; 2023) that project the whole input sequence onto a compact latent space, we only replace $m < t_c$ CoT tokens with their latent abstractions, leaving the remaining tokens unchanged. We delimit the latent tokens by injecting a special `<boLatent>` and `<eoLatent>` tokens to encapsulate them.

**Left-to-Right (AR) Replacement.** We replace the leftmost $m$ tokens of $C$, rather than subsampling tokens at different locations.

**Mixing Samples with Varying Values of** $m$. For fine-tuning an existing LLM on the reasoning dataset with latent tokens, one remarkable challenge is to deal with the extended vocabulary. As the LLM is pretrained with trillions of tokens, it is very hard for it to quickly adapt to tokens (and corresponding embeddings) beyond the original vocabulary. Previous works that aim to replace or eliminate CoT tokens (Deng et al., 2024; Hao et al., 2024) employ a multistage curriculum training approach, where those operations are gradually applied to the entire input sequence. In the context of our approach, this means we increase the values of $m$ in each stage until it reaches a pre-set cap value. However, such training procedure is complex and compu-

tationally inefficient, where dedicated optimization tuning is needed. In this work, we employ a simple single stage training approach where the value of $m$ is randomly set for each sample. Surprisingly, this not only makes our training more efficient, but also leads to enhanced performance.

Note that we use a VQVAE with a size of 50M, adding minimal parameter overhead. In addition, it is used only once during data preparation (to convert training data into discrete latent code), not during LLM training or inference. During inference, the LLM directly generates latent tokens without any use of VQVAE.

## 4 Experiments

We empirically evaluate our approach on two categories of benchmarks:

**(1)** Synthetic datasets including the Keys-Finding Maze, ProntoQA (Saparov & He, 2022), and ProsQA (Hao et al., 2024), where we pretrain T5 or GPT-2 models from scratch using the method in Section 3;

**(2)** Real-world mathematic reasoning problems, where we fine-tune Llama models (Dubey et al., 2024) on the MetaMathQA (Yu et al., 2023) or the Dart-MATH (Tong et al., 2024) dataset, and then test on in-domain datasets Math and GSM-8K, along with out-of-domain datasets including Fresh-Gaokao-Math-2023, DeepMind-Math, College-Math, OlympiaBench-Math, and TheoremQA.

The detailed setup is introduced in Section 4.1.

We compare our approach to the following baselines:

**Sol-Only**: the model is trained with samples that only contains questions and solutions, without any reasoning steps;

**CoT**: the model is trained with samples with complete CoT tokens;

**iCoT** (Deng et al., 2024): a method that utilizes curriculum learning to gradually eliminate the need of CoT tokens in reasoning;

**Pause Token** (Goyal et al., 2023): a method that injects a learnable `pause` token into the sample during training, in order to offer extra computation before giving out the final answer.

### 4.1 Benchmarks

#### 4.1.1 SYNTHETIC BENCHMARKS

**Keys-Finding Maze** is a complex navigation environment designed to evaluate an agent's planning capabilities. The agent is randomly positioned within a maze comprising 4 $3 \times 3$ interconnected rooms, with the objective of reaching a randomly placed goal destination. To successfully reach

the destination, the agent must collect keys (designated with green, red, and blue colors) that correspond to matching colored doors. These keys are randomly distributed among the rooms, requiring the agent to develop sophisticated planning strategies for key acquisition and door traversal. The agent is only allowed to take one key at a time. This environment poses a substantial cognitive challenge, as the agent must identify which keys are necessary for reaching the destination, and optimize the order of key collection and door unlocking to establish the most efficient path to the goal. Following Lehnert et al. (2024); Su et al. (2024), we generate intermediate search traces using the nondeterministic A* algorithm (Hart et al., 1968). The dataset contains 100k training samples. See Appendix A.2 for more information and graphical illustrations.

**ProntoQA** (Saparov & He, 2022) is a dataset consists of 9000 logical reasoning problems derived from ontologies - formal representations of relationships between concepts. Each problem in the dataset is constructed to have exactly one correct proof or reasoning path. One distinctive feature of this dataset is its consistent grammatical and logical structure, which enables researchers to systematically analyze and evaluate how LLMs approach reasoning tasks.

**ProsQA** (Hao et al., 2024) is a more difficult benchmark building on top of ProntoQA. It contains 17,886 logical problems curated by randomly generated directed acyclic graphs. It has larger size of distracting reasoning paths in the ontology, and thus require more complex reasoning and planning capabilities.

### 4.1.2 MATHEMATICAL REASONING

We fine-tune pretrained LLMs using the MetaMathQA (Yu et al., 2023) or the Dart-MATH (Tong et al., 2024) dataset. MetaMathQA is a curated dataset that augments the existing `Math` (Hendrycks et al.) and `GSM8K` (Cobbe et al., 2021b) datasets by various ways of question bootstrapping, such as (i) rephrasing the question and generating the reasoning path; (ii) generating backward questions, self-verification questions, FOBAR questions (Jiang et al., 2024), etc. This dataset contains 395k samples in total, where 155k samples are bootstrapped from `Math` and the remaining 240k come from `GSM8K`. We rerun the MetaMath data pipeline by using Llama-3.1-405B-Inst to generate the response. Dart-MATH (Tong et al., 2024) also synthesizes responses for questions in `Math` and `GSM8K`, with the focus on difficult questions via difficulty-aware rejection tuning. For evaluation, we test the models on the original `Math` and `GSM8K` datasets, which are in-domain, and also the following out-of-domain benchmarks:

- **College-Math** (Tang et al., 2024) consists of 2818 college-level math problems taken from 9 textbooks. These problems cover over 7 different areas such as linear algebra,

differential equations, and so on. They are designed to evaluate how well the language model can handle complicated mathematical reasoning problems in different field of study.

- **DeepMind-Math** (Saxton et al., 2019) consists of 1000 problems based on the national school math curriculum for students up to 16 years old. It examines the basic mathematics and reasoning skills across different topics.

- **OlympiaBench-Math** (He et al., 2024) is a text-only English subset of Olympiad-Bench focusing on advanced level mathematical reasoning. It contains 675 highly difficult math problems from competitions.

- **TheoremQA** (Chen et al., 2023) contains 800 problems focuses on solving problems in STEM fields (such as math, physics, and engineering) using mathematical theorems.

- **Fresh-Gaokao-Math-2023** (Tang et al., 2024) contains 30 math questions coming from Gaokao, or the National College Entrance Examination, which is a national standardized test that plays a crucial role in the college admissions process.

### 4.2 Main Results

We employ a consistent strategy for training VQ-VAE and replacing CoT tokens with latent discrete codes across all our experiments, as outlined below. The specific model architecture and key hyperparameters used for LLM training are presented alongside the results for each category of benchmarks. All the other details are deferred to Appendix A.

**VQ-VAE Training** For each benchmark, we train a VQ-VAE for 100k steps using the Adam optimizer, with learning rate $10^{-5}$ and batch size 32. We use a codebook of size 1024 and compress every chunk of $L = 16$ tokens into a single latent token (i.e., the compression rate $r = 16$).

**Randomized Latent Code Replacement** We introduce a stochastic procedure for partially replacing CoT tokens with latent codes. Specifically, we define a set of predetermined numbers $\mathcal{M} = \{0, 72, 128, 160, 192, 224, 256\}$, which are all multipliers of $L = 16$. For each training example, we first sample $m_{\max} \in \mathcal{M}$ then sample an integer $m \in [0, 16, 32, \ldots, m_{\max}]$ uniformly at random. The first $m$ CoT tokens are replaced by their corresponding latent discrete codes, while the later ones remain as raw text. This stochastic replacement mechanism exposes the model to a wide range of latent-text mixtures, enabling it to effectively learn from varying degrees of latent abstraction.

### 4.2.1 SYNTHETIC BENCHMARKS

**Hyperparameters and Evaluation Metric** For our experiments on the ProntoQA and ProsQA datasets, we fine-tune

| Model | Keys-Finding Maze | | ProntoQA | | ProsQA | |
|---|---|---|---|---|---|---|
| | 1-Feasible-10 (%) | Num. Tokens | Accuracy | Num. Tokens | Accuracy | Num. Tokens |
| Sol-Only | 3 | 645 | 93.8 | 3.0 | 76.7 | 8.2 |
| CoT | 43 | 1312.0 | 98.8 | 92.5 | 77.5 | 49.4 |
| **Latent (ours)** | **62.8 (↑ +19.8)** | 374.6 | **100 (↑ +1.2)** | 7.7 | **96.2 (↑ +18.7)** | 10.9 |

Table 4.1: Our latent approach surpasses the other baselines on Keys-Finding Maze, ProntoQA and ProsQA with a large margin . We use top-$k$ ($k = 10$) decoding for Keys-Finding Maze and greedy decoding for ProntoQA and ProsQA. In terms of token efficiency, our latent approach also generates much shorter reasoning traces than the CoT baseline, closely tracking or even outperforming the Sol-Only approach. **Bold: best results**. Underline: second best results. (↑ +Performance gain compared with the second best result.)

| Model | | In-Domain | | Out-of-Domain | | | | | Average |
|---|---|---|---|---|---|---|---|---|---|
| | | Math | GSM8K | Gaokao-Math-2023 | DM-Math | College-Math | Olympia-Math | TheoremQA | All Datasets |
| | Sol-Only | 4.7 | 6.8 | 0.0 | 10.4 | 5.3 | 1.3 | 3.9 | 4.6 |
| | CoT | 10.5 | 42.7 | **10.0** | 3.4 | 17.1 | 1.5 | 9.8 | 14.1 |
| **Llama-3.2-1B** | iCoT | 8.2 | 10.5 | 3.3 | 11.3 | 7.6 | **2.1** | 10.7 | 7.7 |
| | Pause Token | 5.1 | 5.3 | 2.0 | 1.4 | 0.5 | 0.0 | 0.6 | 2.1 |
| | **Latent (ours)** | **14.7 (↑ +4.2)** | **48.7 (↑ +6)** | **10.0** | **14.6 (↑ +3.3)** | **20.5 (↑ +3.4)** | 1.8 | **11.3 (↑ +0.6)** | **17.8 (↑ +3.7)** |
| | Sol-Only | 6.1 | 8.1 | 3.3 | 14.0 | 7.0 | 1.8 | 6.8 | 6.7 |
| | CoT | 21.9 | 69.7 | 16.7 | **27.3** | 30.9 | 2.2 | 11.6 | 25.2 |
| **Llama-3.2-3B** | iCoT | 12.6 | 17.3 | 3.3 | 16.0 | 14.2 | **4.9** | **13.9** | 11.7 |
| | Pause Token | 25.2 | 53.7 | 4.1 | 7.4 | 11.8 | 0.7 | 1.0 | 14.8 |
| | **Latent (ours)** | **26.1 (↑ +4.2)** | **73.8 (↑ +4.1)** | **23.3 (↑ +6.6)** | 27.1 | **32.9 (↑ +2)** | 4.2 | 13.5 | **28.1 (↑ +2.9)** |
| | Sol-Only | 11.5 | 11.8 | 3.3 | 17.4 | 13.0 | 3.8 | 6.7 | 9.6 |
| | CoT | 32.9 | 80.1 | 16.7 | 39.3 | 41.9 | 7.3 | 15.8 | 33.4 |
| **Llama-3.1-8B** | iCoT | 17.8 | 29.6 | 16.7 | 20.3 | 21.3 | 7.6 | 14.8 | 18.3 |
| | Pause Token | **39.6** | 79.5 | 6.1 | 25.4 | 25.1 | 1.3 | 4.0 | 25.9 |
| | **Latent (ours)** | 37.2 | **84.1 (↑ +4.0)** | **30.0 (↑ +13.3)** | **41.3 (↑ +2)** | **44.0 (↑ +2.1)** | **10.2 (↑ +2.6)** | **18.4 (↑ +2.6)** | **37.9 (↑ +4.5)** |

Table 4.2: Our latent approach outperforms the baselines on various types of mathematical reasoning benchmarks. The models are fine-tuned on the MetaMathQA (Yu et al., 2023) dataset. The Math and GSM8K are in-domain datasets since they are used to generate MetaMathQA, while the others are out-of-domain. **Bold: best results**. Underscore: second best results. ↑ +: Performance gain compared with the second best result.

the pretrained GPT-2 model (Radford et al., 2019) for 16k steps, where we use a learning rate of $10^{-4}$ with linear warmup for 100 steps, and the batch size is set to 128. To evaluate the models, we use greedy decoding and check the exact match with the ground truth.

For Keys-Finding Maze, due to its specific vocabulary, we trained a T5 model (Raffel et al., 2020) from scratch for 100k steps with a learning rate of $7.5 \times 10^{-4}$ and a batch size of 1024. We evaluate the models by the *1-Feasible-10* metric. Namely, for each evaluation task, we randomly sample 10 responses with top-$k$ ($k$=10) decoding and check if any of them is feasible and reaches the goal location.

**Results** As shown in Table 4.1, our latent approach performs better than the baselines for both the Keys-Finding Maze and ProntoQA tasks. Notably, the absolute improvement is 15% for the Keys-Finding Maze problem, and we reach 100% accuracy on the relatively easy ProntoQA dataset. For the more difficult ProsQA, the CoT baseline only obtains 77.5% accuracy, the latent approach achieves 17.5% performance gain.

### 4.2.2 MATHEMATICAL REASONING

**Hyperparameters and Evaluation Metrics** We considered 3 different sizes of LLMs from the LLaMa herd: Llama-3.2-1B, Llama-3.2-3B and Llama-3.1-8B models. For all the models, we fine-tune them on the MetaMathQA dataset for 1 epoch. To maximize training efficiency, we use a batch size of 32 with a sequence packing of 4096. We experiment with different learning rates $10^{-5}, 2.5 \times 10^{-5}, 5 \times 10^{-5}, 10^{-4}$ and select the one with the lowest validation error. The final choices are $10^{-5}$ for the 8B model and $2.5 \times 10^{-5}$ for the others. For all the experiments, we use greedy decoding for evaluation.

**Accuracy Comparison** Table 4.2 presents the results. Our latent approach consistently outperforms all the baselines across nearly all the tasks, for models of different sizes. For tasks on which we do not observe improvement, our approach is also comparable to the best performance. The gains are more pronounced in specific datasets such as Gaokao-Math-2023. On average, we are observing a $+5.3$ points improvement for the 8B model, $+2.9$ points improvement for the 3B model, and $+3.7$ points improvement for

| Model | | In-Domain (# of tokens) | | Out-of-Domain (# of tokens) | | | | | Average |
|---|---|---|---|---|---|---|---|---|---|
| | | Math | GSM8K | Gaokao-Math-2023 | DM-Math | College-Math | Olympia-Math | TheoremQA | All Datasets |
| **Llama-3.2-1B** | Sol-Only | 4.7 | 6.8 | 0.0 | 10.4 | 5.3 | 1.3 | 3.9 | 4.6 |
| | CoT | 646.1 | 190.3 | 842.3 | 578.7 | 505.6 | 1087.0 | 736.5 | 655.2 |
| | iCoT | 328.4 | 39.8 | 354.0 | 170.8 | 278.7 | 839.4 | 575.4 | 369.5 |
| | Pause Token | 638.8 | 176.4 | 416.1 | 579.9 | 193.8 | 471.9 | 988.1 | 495 |
| | **Latent (ours)** | 501.6 ($\downarrow$ **-22%**) | 181.3 ($\downarrow$ **-5%**) | 760.5 ($\downarrow$ **-11%**) | 380.1 ($\downarrow$ **-34%**) | 387.3 ($\downarrow$ **-23%**) | 840.0 ($\downarrow$ **-22%**) | 575.5 ($\downarrow$ **-22%**) | 518 ($\downarrow$ **-21%**) |
| **Llama-3.2-3B** | Sol-Only | 6.1 | 8.1 | 3.3 | 14.0 | 7.0 | 1.8 | 6.8 | 6.7 |
| | CoT | 649.9 | 212.1 | 823.3 | 392.8 | 495.9 | 1166.7 | 759.6 | 642.9 |
| | iCoT | 344.4 | 60.7 | 564.0 | 154.3 | 224.9 | 697.6 | 363.6 | 344.2 |
| | Pause Token | 307.9 | 162.3 | 108.9 | 251.5 | 500.96 | 959.5 | 212.8 | 354.7 |
| | **Latent (ours)** | 516.7 ($\downarrow$ **-20%**) | 198.8 ($\downarrow$ **-6%**) | 618.5 ($\downarrow$ **-25%**) | 340.0 ($\downarrow$ **-13%**) | 418.0 ($\downarrow$ **-16%**) | 832.8 ($\downarrow$ **-29%**) | 670.2 ($\downarrow$ **-12%**) | 513.6 ($\downarrow$ **-20%**) |
| **Llama-3.1-8B** | Sol-Only | 11.5 | 11.8 | 3.3 | 17.4 | 13.0 | 3.8 | 6.7 | 9.6 |
| | CoT | 624.3 | 209.5 | 555.9 | 321.8 | 474.3 | 1103.3 | 760.1 | 578.5 |
| | iCoT | 403.5 | 67.3 | 444.8 | 137.0 | 257.1 | 797.1 | 430.9 | 362.5 |
| | Pause Token | 469.4 | 119.0 | 752.6 | 413.4 | 357.3 | 648.2 | 600.1 | 480 |
| | **Latent (ours)** | 571.9 ($\downarrow$ **-9 %**) | 193.9 ($\downarrow$ **-8 %**) | 545.8 ($\downarrow$ **-2 %**) | 292.1 ($\downarrow$ **-10%**) | 440.3 ($\downarrow$ **-8%**) | 913.7 ($\downarrow$ **-17 %**) | 637.2 ($\downarrow$ **-16 %**) | 513.7 ($\downarrow$ **-10%**) |

Table 4.3: The average number of tokens in the generated responses. Compared with the CoT baseline, our latent approach achieves an 17% reduction in response length on average, while surpassing it in final performance according to Table 4.2. The iCoT method generates shorter responses than our approach, yet performs significantly worse, see Table 4.2. $\downarrow$ -: Trace length reduction rate compared with CoT.

| Model | | In-Domain | | Out-of-Domain | | | | | Average |
|---|---|---|---|---|---|---|---|---|---|
| | | math | GSM8K | Fresh-Gaokao-Math-2023 | DeepMind-Mathematics | College-Math | Olympia-Math | TheoremQA | All Datasets |
| **Llama-3.2-1B** | All-Replace | 6.7 | 4.2 | 0.0 | 11.8 | 6.0 | 2.1 | 8.5 | 5.6 |
| | Curriculum-Replace | 7.1 | 9.8 | 3.3 | 13.0 | 7.9 | **2.4** | 10.5 | 7.8 |
| | Poisson-Replace | 13.9 | **49.5** | 10.0 | 12.2 | 18.9 | 2.3 | 9.0 | 15.1 |
| | **Latent-AR (ours)** | **14.7** | 48.7 | **10.0** | **14.6** | **20.5** | 1.8 | **11.3** | **17.8** |
| **Llama-3.2-3B** | All-Replace | 10.7 | 12.8 | 10.0 | 19.4 | 12.8 | **5.3** | 11.8 | 11.8 |
| | Curriculum-Replace | 10.2 | 14.9 | 3.3 | 16.8 | 12.9 | 3.9 | **14.4** | 10.9 |
| | Poisson-Replace | 23.6 | 65.9 | 13.3 | 17.9 | 28.9 | 2.9 | 11.2 | 20.5 |
| | **Latent (ours)** | **26.1** | **73.8** | **23.3** | **27.1** | **32.9** | 4.2 | 13.5 | **28.1** |
| **Llama-3.1-8B** | All-Replace | 15.7 | 19.9 | 6.7 | 21.1 | 19.5 | 5.0 | 17.5 | 15.0 |
| | Curriculum-Replace | 14.6 | 23.1 | 13.3 | 20.3 | 18.7 | 3.9 | 16.6 | 15.8 |
| | Possion-Replace | **37.9** | 83.6 | 16.6 | **42.7** | **44.7** | 9.9 | **19.1** | 36.3 |
| | **Latent (ours)** | 37.2 | **84.1** | **30.0** | 41.3 | 44.0 | **10.2** | 18.4 | **37.9** |

Table 4.4: Our latent token replacement strategy significantly outperforms the alternative choices: All-Replace (where all the textual CoT tokens are replaced by latent tokens at once), Curriculum-Replace (where we gradually replace the text tokens for the entire CoT subsequence by latent tokens over the course of training) and Poisson-Replace (where individual chunks of text tokens are replaced with probabilities 0.5).

the 1B model.

**Tokens Efficiency Comparison** Alongside the accuracy, we also report the number of tokens contained in the generated responses in Table 4.3, which is the dominating factor of the inference efficiency. Our first observation is that for all the approaches, the model size has little influence on the length of generated responses. Overall, the CoT method outputs the longest responses, while the Sol-Only method outputs the least number of tokens, since it is trained to generate the answer directly. The iCoT method generates short responses as well (42.8% reduction compared to CoT), as the CoT data has been iteratively eliminated in its training procedure. However, this comes at the cost of significantly degraded model performance compared with CoT, as shown in Table 4.2. Our latent approach shows an average 17% reduction in token numbers compared with CoT while surpassing it in prediction accuracy.

### 4.3 Ablation & Understanding Studies

**Replacement Strategies** Our latent approach partially replaces the leftmost $m$ CoT tokens, where the value of $m$ varies for each sample. We call such replacement strategies **AR-Replace**. Here we consider three alternative strategies:

(1) **All-Replace**: all the text CoT tokens are replaced by the latent tokens.

(2) **Curriculum-Replace**: the entire CoT subsequence are gradually replaced over the course of training, similar to the training procedure used by iCoT and CO-CONUT (Hao et al., 2024). We train the model for 8 epochs. Starting from the original dataset, in each epoch we construct a new training dataset whether we further replace the leftmost 16 textual CoT tokens by a discrete latent token.

(3) **Poisson-Replace**: instead of replacing tokens from

left to right, we conduct a *Poisson sampling* process to select CoT tokens to be replaced: we split the reasoning traces into chunks consisting of 16 consecutive text tokens, where each chunk is randomly replaced by the latent token with probability 0.5.

Table 4.4 reports the results. Our **AR-Replace** strategy demonstrate strong performance, outperforming the other two strategies with large performance gap. Our intuition is as follows. When all the textual tokens are removed, the model struggles to align the latent tokens with the linguistic and semantic structures it learned during pretraining.

In contrast, partial replacement offers the model a bridge connecting text and latent spaces: the remaining text tokens serve as anchors, helping the model interpret and integrate the latent representations more effectively. Interestingly, the curriculum learning strategy is bridging the two spaces very well, where **All-Replace** and **Curriculum-Replace** exhibit similar performance. This is similar to our observation that iCoT performs remarkably worse than CoT for mathematical reasoning problems. **Poisson-Replace** demonstrates performance marginally worse to our **AR-Replace** strategy on the 1B and 8B models, but significantly worse on the 3B model. Our intuition is that having a fix pattern of replacement (starting from the beginning and left to right) is always easier for the model to learn. This might be due to the limited finetuning dataset size and model capacity.

**Attention Weights Analysis** To understand the reason why injecting latent tokens enhanced the model's reasoning performance, we randomly selected two questions from the Math and College-Math dataset and generate responses, then analyze the attention weights over the input prompt tokens:

1. What is the positive difference
   between $120%$ of 30 and $130%$ of
   20?

2. Mark has $50 in his bank account.
   He earns $10 per day at his work.
   If he wants to buy a bike that
   costs $300, how many days does Mark
   have to save his money?

Specifically, we take the last attention layer, compute the average attention weights over different attention heads and show its relative intensity over the prompt tokens[2]. We com-

_________________

[2]We first compute the average attention weights across multiple heads. This gives us a single lower triangular matrix. Then, we take the column sum of this matrix to get an aggregated attention weights for each token. Last, we normalize the weights by their average to obtain the relative intensity. A one line pseudocode is: `column_sum(avg(attention_matrices)) / avg(column_sum(avg(attention_matrices)))`.

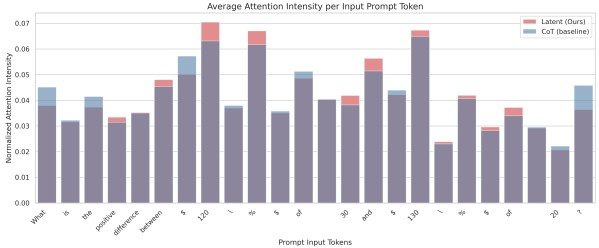

(a) Prompt: `What is the positive difference between $120%$ of 30 and $130%$ of 20?`

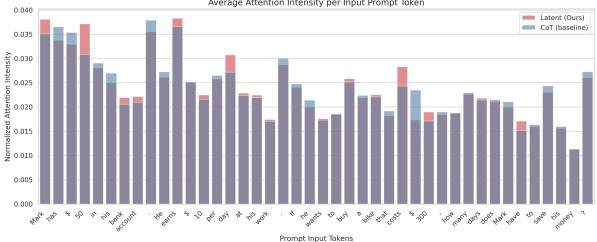

(b) Prompt: `Mark has $50 in his bank account. He earns $10 per day at his work. If he wants to buy a bike that costs $300, how many days does Mark have to save his money?`

Figure 4.1: Comparing with the CoT model, our latent approach have high attention weights on numbers and text tokens representing mathematical operations.

pare the averaged attention weights of our model with the CoT model in Figure 4.1. Interestingly, our model learns to grasp a stronger attention to numbers and words representing mathematical operations. Both Figure 4.1a and Figure 4.1b show that the latent model focus more on the numbers, such as `120`, `30`, and `130` for the first question. For the second question, our latent model shows a larger attention weights on numbers including `50`, `10`, and `300`, and also tokens semantically related to mathematical operations such as `earns` (means addition) and `cost` (means subtraction). This suggests that, by partially compressing the reasoning trace into a mix of latent and text tokens, we allow the model to effectively focus on important tokens that build the internal logical flow. See Appendix C.1 for the exact response generated by our approach and the CoT baseline.

## 4.4 Ablations on the Latent $r$ parameters

Throughout this paper we have been using $r$ (or the compression ratio) to be 16, in this section, we will be ablating how would $r$ affects the performance of the downstream Math tasks if we vary this parameter.

To this end, we vary this parameter on the Llama-3.2-3B model. Our result is summarized in Table.4.5. A graphical illustration is shown in Figure.4.2. A key takeaway is that

Table 4.5: The Table blow illustrates a clear trend on Llama-3.2-3B model where increasing the compression rate reduces the number of generated tokens due to higher data compression. Notably, a compression rate of 2 shows some improvements over the CoT baseline. Furthermore, there appears to be an optimal 'sweet spot' where the data is neither overly compressed (rate = 32) nor minimally compressed (rate = 2), optimizing both efficiency and accuracy.

| | Model | **In-Domain** | | **Out-of-Domain** | | | | | **Average** |
|---|---|---|---|---|---|---|---|---|---|
| | | math | GSM8K | Fresh-Gaokao-Math-2023 | DeepMind-Mathematics | College-Math | Olympia-Math | TheoremQA | All Datasets |
| **Llama-3.2-3B** | CoT (baseline) (Acc.) | 21.9 | 69.7 | 16.7 | 27.3 | 30.9 | 2.2 | 11.6 | 25.2 |
| | # of Tokens | 649.9 | 212.1 | 823.3 | 392.8 | 495.9 | 1166.7 | 759.6 | 642.9 |
| | Latent-$r = 2$ (Acc.) | 24.3 | 71.7 | 16.7 | 25.4 | 32.0 | 4.7 | 14.8 | 27.08 |
| | # of Tokens | 586.0 | 207.6 | 739.6 | 415.3 | 471 | 1036 | 714 | 595.6 |
| | Latent $r = 16$ (Acc.) | 26.1 | 73.8 | 23.3 | 27.1 | 32.9 | 4.2 | 13.5 | 28.1 |
| | # of Tokens | 516.7 | 198.8 | 618.5 | 340.0 | 418.0 | 832.8 | 670.2 | 513.6 |
| | Latent $r = 32$ (Acc.) | 25.2 | 71.5 | 23.3 | 26.3 | 33.3 | 4.9 | 14.1 | 27.9 |
| | # of Tokens | 496.5 | 183.3 | 577.3 | 311.0 | 395.2 | 821.0 | 585.6 | 481.4 |

our latent approach comes out ahead of the CoT baseline for all $r$ settings in terms of fewer tokens (better efficiency) and higher accuracy. This is a strong signal that the shift to a latent representation itself is fundamentally beneficial. In addition, we see that when the $r$ (compression ratio) increases, we expect each latent token to encode more information (higher compression). As a result, we see that, on average, the number of tokens reduces as $r$ increases. However, in terms of the accuracy metric, we see that the model increases initially from 25.2 (overall accuracy) to 27.1 (when $r = 2$). It further boosts up to 28.1 at $r = 16$, and then it goes down to 27.9 when $r = 32$. This indicates a sweet spot that $r = 16$, it is neither overly-compressed (which implies information loss), nor under-compressed (which implies information is not encoded abstractly enough). This study indicates an interesting trade-off between accuracy and tokens efficiency in our latent approach. So, $r = 16$ appears to strike an optimal balance between compact representation and the preservation of task-critical information.

## 4.5 Additional Examples and Interpretability Result

We provide 4 additional example responses for questions in the Math and TheoremQA datasets in Appendix D. In Appendix F, we compare all the approaches when the model is trained on the DART-MATH (Tong et al., 2024) dataset, where similar trends are observed.

We also provide interpretable examples in the Appendix E.

## 5 Conclusion

We present a novel approach to improve the reasoning capabilities of LLMs, by compressing the initial steps of the reasoning traces using discrete latent tokens obtained from VQ-VAE. By integrating both abstract representation and

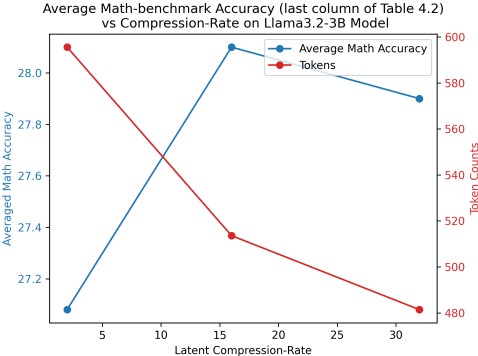

Figure 4.2: A graphical illustration of the compression rate $r$ trade-off between the accuracy and the token efficiency on the Llama-3.2-3B model.

textual details of the reasoning process into training, our approach enables LLMs to capture essential reasoning information with improved token efficiency. Furthermore, by randomizing the number of text tokens to be compressed during training, we unlock fast adaptation to unseen latent tokens. Our comprehensive evaluation demonstrates the effectiveness across multiple domains, outperforming standard methods that rely on complete textual reasoning traces.

## Impact Statement

This paper presents a method to enhance the reasoning capability of Large Language Models (LLMs) by combining latent and text tokens in the reasoning trace. In terms of society impact, while reasoning with (opaque) latent tokens may trigger safety concerns, our approach provides a VQ-VAE decoder that can decode the latent tokens into human readable format, mitigating such concerns.

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

# A Experiment Details

## A.1 VQ-VAE Model Details

The codebook size $|\mathcal{E}|$ is 64 for ProntoQA and ProsQA, 512 for the Keys-Finding Maze, and 1024 for math reasoning problems. For both encoder $f_{enc}$ and decoder $f_{dec}$, we use a 2-layer transformer with 4 heads, where the embedding size is 512 and the block size is 512. We set the max sequence to be 2048 for the synthetic dataset experiments and 256 for the math reasoning experiments.

## A.2 Keys-Finding Maze

### A.2.1 ENVIRONMENT DETAILS

In this section, we introduce our synthetic keys-finding maze environment. Figure A.1 shows an example maze that consists of $m \times m$ rooms, where the size of each room is $n \times n$ ($m = 3$ and $n = 5$). The goal of the agent (represented by the black circle) is to reach the gold diamond using the minimum number of steps. The agent cannot cross the wall. Also, there are three doors (represented by squares) of different colors (i.e., red, green, and blue) which are closed initially. The agent have to pick up keys to open the door in the same color. Note that the agent can not carry more than one key at the same time.

Figure A.2 shows an example optimal trajectory of the maze in Figure A.1. The agent first picks up the blue key and opens the blue door to obtain the red key. Then the agent navigates to the red door and opens it. Finally the agent is able to reach the objective.

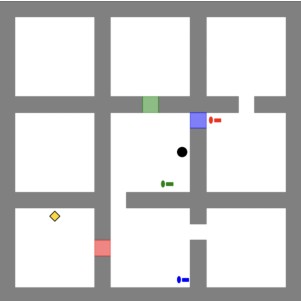

Figure A.1: An example of the keys-finding maze environment.

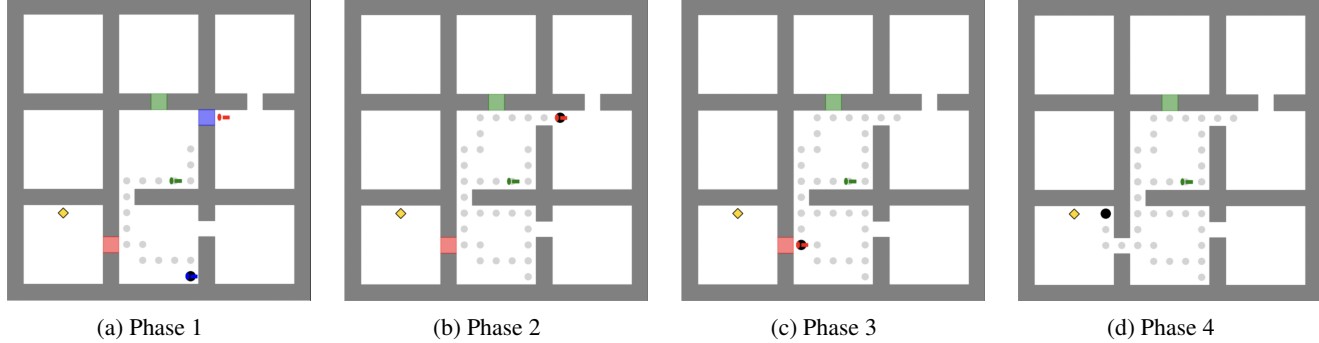

|          (a) Phase 1          |          (b) Phase 2          |          (c) Phase 3          |          (d) Phase 4          |

Figure A.2: An (optimal) trajectory of the maze in Figure A.1. Phase 1: the agent picks up the blue key; Phase 2: the agent opens the blue door to obtain the red key; Phase 3: the agent carries the red key to the red door; Phase 4: the agent opens the red door and reaches the objective.

### A.2.2 DATASET DETAILS

Our dataset consists of 100k training data points, 500 validation data points, and 300 data points for testing. For each data point, the structure of the prompt and response is as follows:

- [Prompt]:
  **maze_size**: $M \times M$
  **agent**: $(x_{a0}, y_{a0})$,
  **walls**: $(x_1, y_1), (x_2, y_2), \ldots$
  **objective**: $(x_o, y_o)$,
  **keys**: [red_key]: $(x_{rk}, y_{rk}), \ldots$
  **doors**: [red_door]: $(x_{rd}, y_{rd}), \ldots$

- [Response]:
  create-node $(x_{a1}, y_{a1}, f_{a1}, h_{a1})$,
  create-node $(x_{a2}, y_{a2}, f_{a2}, h_{a2})$,
  ...
  agent $(x_{a1}, y_{a1}), (x_{a2}, y_{a2}), \ldots, (x_{aT}, y_{aT})$,

Below, we show the prompt and response for an example training data pint.

> **Prompt**
>
> initial_state: maze_size: 19x19 wall: (0,0), (0,1), (0,2), (0,3), (0,4), (0,5), (0,6), (0,7), (0,8), (0,9), (0,10), (0,11), (0,12), (0,13), (0,14), (0,15), (0,16), (0,17), (0,18), (1,0), (1,6), (1,12), (1,18), (2,0), (2,6), (2,12), (2,18), (3,0), (3,6), (3,12), (3,18), (4,0), (4,6), (4,12), (4,18), (5,0), (5,6), (5,12), (5,18), (6,0), (6,1), (6,3), (6,4), (6,5), (6,6), (6,7), (6,8), (6,9), (6,10), (6,11), (6,12), (6,13), (6,14), (6,15), (6,16), (6,17), (6,18), (7,0), (7,12), (7,18), (8,0), (8,6), (8,12), (8,18), (9,0), (9,6), (9,12), (9,18), (10,0), (10,6), (10,12), (10,18), (11,0), (11,6), (11,12), (11,18), (12,0), (12,1), (12,2), (12,3), (12,4), (12,6), (12,8), (12,9), (12,10), (12,11), (12,12), (12,13), (12,14), (12,15), (12,16), (12,17), (12,18), (13,0), (13,12), (13,18), (14,0), (14,6), (14,12), (14,18), (15,0), (15,6), (15,12), (15,18), (16,0), (16,6), (16,12), (16,18), (17,0), (17,6), (17,12), (17,18), (18,0), (18,1), (18,2), (18,3), (18,4), (18,5), (18,6), (18,7), (18,8), (18,9), (18,10), (18,11), (18,12), (18,13), (18,14), (18,15), (18,16), (18,17), (18,18), agent: (11,5), objective: (1,1), [red_key]: (16,1), [green_key]: (16,9), [blue_key]: (7,2), [red_door]: (12,5), [green_door]: (6,2), [blue_door]: (12,7)

---

Response (CoT and Answer)

create-node: (11,5), heuristics: 14, cost: 0 create-node: (11,4), heuristics: 13, cost: 1 create-node: (11,3), heuristics: 12, cost: 2 create-node: (10,3), heuristics: 11, cost: 3 create-node: (9,3), heuristics: 10, cost: 4 create-node: (9,2), heuristics: 9, cost: 5 create-node: (8,2), heuristics: 8, cost: 6 create-node: (7,2), heuristics: 7, cost: 7 create-node: (7,2), heuristics: 7, cost: 8 create-node: (7,3), heuristics: 8, cost: 9 create-node: (7,4), heuristics: 9, cost: 10 create-node: (7,5), heuristics: 10, cost: 11 create-node: (7,6), heuristics: 11, cost: 12 create-node: (7,7), heuristics: 12, cost: 13 create-node: (8,7), heuristics: 13, cost: 14 create-node: (9,7), heuristics: 14, cost: 15 create-node: (10,7), heuristics: 15, cost: 16 create-node: (11,7), heuristics: 16, cost: 17 create-node: (11,7), heuristics: 16, cost: 18 create-node: (12,7), heuristics: 17, cost: 19 create-node: (13,7), heuristics: 18, cost: 20 create-node: (14,7), heuristics: 19, cost: 21 create-node: (14,8), heuristics: 20, cost: 22 create-node: (15,8), heuristics: 21, cost: 23 create-node: (15,9), heuristics: 22, cost: 24 create-node: (16,9), heuristics: 23, cost: 25 create-node: (16,9), heuristics: 23, cost: 26 create-node: (16,8), heuristics: 22, cost: 27 create-node: (16,7), heuristics: 21, cost: 28 create-node: (15,7), heuristics: 20, cost: 29 create-node: (14,7), heuristics: 19, cost: 30 create-node: (13,7), heuristics: 18, cost: 31 create-node: (12,7), heuristics: 17, cost: 32 create-node: (11,7), heuristics: 16, cost: 33 create-node: (10,7), heuristics: 15, cost: 34 create-node: (9,7), heuristics: 14, cost: 35 create-node: (8,7), heuristics: 13, cost: 36 create-node: (7,7), heuristics: 12, cost: 37 create-node: (7,6), heuristics: 11, cost: 38 create-node: (7,5), heuristics: 10, cost: 39 create-node: (7,4), heuristics: 9, cost: 40 create-node: (7,3), heuristics: 8, cost: 41 create-node: (7,2), heuristics: 7, cost: 42 create-node: (7,2), heuristics: 7, cost: 43 create-node: (6,2), heuristics: 6, cost: 44 create-node: (5,2), heuristics: 5, cost: 45 create-node: (5,1), heuristics: 4, cost: 46 create-node: (4,1), heuristics: 3, cost: 47 create-node: (3,1), heuristics: 2, cost: 48 create-node: (2,1), heuristics: 1, cost: 49 create-node: (1,1), heuristics: 0, cost: 50 agent: (11,5), current_key: none, remaining_key: [red_key]: [green_key]: [blue_key]: remaining_door: [red_door]: [green_door]: [blue_door]: agent: (11,4), current_key: none, remaining_key: [red_key]: [green_key]: [blue_key]: remaining_door: [red_door]: [green_door]: [blue_door]: agent: (11,3), current_key: none, remaining_key: [red_key]: [green_key]: [blue_key]: remaining_door: [red_door]: [green_door]: [blue_door]: agent: (10,3), current_key: none, remaining_key: [red_key]: [green_key]: [blue_key]: remaining_door: [red_door]: [green_door]: [blue_door]: agent: (9,3), current_key: none, remaining_key: [red_key]: [green_key]: [blue_key]: remaining_door: [red_door]: [green_door]: [blue_door]: agent: (9,2), current_key: none, remaining_key: [red_key]: [green_key]: [blue_key]: remaining_door: [red_door]: [green_door]: [blue_door]: agent: (8,2), current_key: none, remaining_key: [red_key]: [green_key]: [blue_key]: remaining_door: [red_door]: [green_door]: [blue_door]: agent: (7,2), current_key: none, remaining_key: [red_key]: [green_key]: [blue_key]: remaining_door: [red_door]: [green_door]: [blue_door]: agent: (7,2), current_key: none, remaining_key: [red_key]: [green_key]: [blue_key]: remaining_door: [red_door]: [green_door]: [blue_door]: agent: (7,3), current_key: [blue_key], remaining_key: [red_key]: [green_key]: remaining_door: [red_door]: [green_door]: [blue_door]: agent: (7,4), current_key: [blue_key], remaining_key: [red_key]: [green_key]: remaining_door: [red_door]: [green_door]: [blue_door]: agent: (7,5), current_key: [blue_key], remaining_key: [red_key]: [green_key]: remaining_door: [red_door]: [green_door]: [blue_door]: agent: (7,6), current_key: [blue_key], remaining_key: [red_key]: [green_key]: remaining_door: [red_door]: [green_door]: [blue_door]: agent: (7,7), current_key: [blue_key], remaining_key: [red_key]: [green_key]: remaining_door: [red_door]: [green_door]: [blue_door]: agent: (8,7), current_key: [blue_key], remaining_key: [red_key]: [green_key]: remaining_door: [red_door]: [green_door]: [blue_door]: agent: (9,7), current_key: [blue_key], remaining_key: [red_key]: [green_key]: remaining_door: [red_door]: [green_door]: [blue_door]: agent: (10,7), current_key: [blue_key], remaining_key: [red_key]: [green_key]: remaining_door: [red_door]: [green_door]: [blue_door]: agent: (11,7), current_key: [blue_key], remaining_key: [red_key]: [green_key]: remaining_door: [red_door]: [green_door]: [blue_door]: agent: (11,7), current_key: none, remaining_key: [red_key]: [green_key]: remaining_door: [red_door]: [green_door]: agent: (12,7), current_key: none, remaining_key: [red_key]: [green_key]: remaining_door: [red_door]: [green_door]: agent: (13,7), current_key: none, remaining_key: [red_key]: [green_key]: remaining_door: [red_door]: [green_door]: agent: (14,7), current_key: none, remaining_key: [red_key]: [green_key]: remaining_door: [red_door]: [green_door]: agent: (14,8), current_key: none, remaining_key: [red_key]: [green_key]: remaining_door: [red_door]: [green_door]: agent: (15,8), current_key: none, remaining_key: [red_key]: [green_key]: remaining_door: [red_door]: [green_door]: agent: (15,9), current_key: none, remaining_key: [red_key]: [green_key]: remaining_door: [red_door]: [green_door]: agent: (16,9), current_key: none, remaining_key: [red_key]: [green_key]: remaining_door: [red_door]: [green_door]: agent: (16,9), current_key: [green_key], remaining_key: [red_key]: remaining_door: [red_door]: [green_door]: agent: (16,8), current_key: [green_key], remaining_key: [red_key]: remaining_door: [red_door]: [green_door]: agent: (16,7), current_key: [green_key], remaining_key: [red_key]: remaining_door: [red_door]: [green_door]: agent: (15,7), current_key: [green_key], remaining_key: [red_key]: remaining_door: [red_door]: [green_door]: agent: (14,7), current_key: [green_key], remaining_key: [red_key]: remaining_door: [red_door]: [green_door]: agent: (13,7), current_key: [green_key], remaining_key: [red_key]: remaining_door: [red_door]: [green_door]: agent: (12,7), current_key: [green_key], remaining_key: [red_key]: remaining_door: [red_door]: [green_door]: agent: (11,7), current_key: [green_key], remaining_key: [red_key]: remaining_door: [red_door]: [green_door]: agent: (10,7), current_key: [green_key], remaining_key: [red_key]: remaining_door: [red_door]: [green_door]: agent: (9,7), current_key: [green_key], remaining_key: [red_key]: remaining_door: [red_door]: [green_door]: agent: (8,7), current_key: [green_key], remaining_key: [red_key]: remaining_door: [red_door]: [green_door]: agent: (7,7), current_key: [green_key], remaining_key: [red_key]: remaining_door: [red_door]: [green_door]: agent: (7,6), current_key: [green_key], remaining_key: [red_key]: remaining_door: [red_door]: [green_door]: agent: (7,5), current_key: [green_key], remaining_key: [red_key]: remaining_door: [red_door]: [green_door]: agent: (7,4), current_key: [green_key], remaining_key: [red_key]: remaining_door: [red_door]: [green_door]: agent: (7,3), current_key: [green_key], remaining_key: [red_key]: remaining_door: [red_door]: [green_door]: agent: (7,2), current_key: [green_key], remaining_key: [red_key]: remaining_door: [red_door]: [green_door]: agent: (7,2), current_key: none, remaining_key: [red_key]: remaining_door: [red_door]: agent: (6,2), current_key: none, remaining_key: [red_key]: remaining_door: [red_door]: agent: (5,2), current_key: none, remaining_key: [red_key]: remaining_door: [red_door]: agent: (5,1), current_key: none, remaining_key: [red_key]: remaining_door: [red_door]: agent: (4,1), current_key: none, remaining_key: [red_key]: remaining_door: [red_door]: agent: (3,1), current_key: none, remaining_key: [red_key]: remaining_door: [red_door]: agent: (2,1), current_key: none, remaining_key: [red_key]: remaining_door: [red_door]: agent: (1,1), current_key: none, remaining_key: [red_key]: remaining_door: [red_door]:

The prompt describes the maze in a structured language. The maze size $M = m(n + 1) + 1$ (e.g., in Figure A.1, the maze size $M = 19$). The positions of walls are $(x_1, y_1), (x_2, y_2), \ldots$, and so on. The position of the agent in time step $t$ is $(x_{at}, y_{at})$, where $t = 0$ corresponds to the initial position The position of the objective is $(x_o, y_o)$, and the position of keys and doors in color $c$ (where $c = r, g, b$) are $(x_{ck}, y_{ck})$ and $(x_{cd}, y_{cd})$, respectively. The response describes an optimal path (i.e., with minimal total times steps $T$) for the agent to reach the objective.

### A.2.3 MODEL DETAILS

Following Su et al. (2024); Lehnert et al. (2024), we employ a similar encode-decoder transformer architecture with rotary embeddings and no drop-out. Our model consisted of 6 layers with 3 attention heads, and the embedding size is 64.

### A.3 ProntoQA and ProsQA

We used the pretrained GPT-2 model which has the following parameters:

| Parameter | Value |
|---|---|
| Number of Layers (Transformer Blocks) | 12 |
| Hidden Size (Embedding Size) | 768 |
| Number of Attention Heads | 12 |
| Vocabulary Size | 50,257 |
| Total Number of Parameters | 117 million |

Table A.1: Hyperparameters of the pretrained GPT-2 model used for ProntoQA and ProsQA.

## A.4 LLM experiments

We use the Llama Cookbook[3] codebase to fine-tune the Llama models.

As described in Section 4.2, we use a batch size of 32 with a sequence packing of 4096. We experiment with different learning rates $10^{-5}, 2.5 \times 10^{-5}, 5 \times 10^{-5}, 10^{-4}$ and select the one with the lowest validation error. The final choices are $10^{-5}$ for Llama-3.2-8B and $2.5 \times 10^{-5}$ for Llama-3.2-1B and Llama-3.2-3B.

## B Notations

Table B.1 summarizes the notations we used throughout the paper.

| | |
|---|---|
| $X = P \oplus C \oplus S$ | input text sample where $\oplus$ means concatenation |
| $P$ | prompt of length $t_p$ |
| $p_i$ | the $i$-th token of prompt (in text) |
| $C$ | reasoning trace of length $t_c$ |
| $c_i$ | the $i$-th token of trace (in text) |
| $S$ | solution of length $t_s$ |
| $s_i$ | the $i$-th token of solution (in text) |
| $Z$ | the complete latent reasoning traces of length $t_z$ |
| $z_i$ | the $i$-th token of latent trace |
| $r = t_c/t_z$ | compression rate |
| $m$ | number of trace tokens to be replaced by latent tokens during training |
| $\widetilde{X}$ | modified input with mixed text and latent tokens |
| $\mathcal{E}$ | codebook of VQ-VAE |
| $e_i$ | the $i$-th vector in the codebook, which corresponds to the $i$-th latent token |
| $d$ | dimension of $e_i$s |
| $\mathcal{V}$ | vocabulary of text tokens |
| $L$ | chunk size |
| $f_{\text{enc}}(\cdot)$ | encodes a chunk of $L$ text tokens to $\frac{L}{r}$ embedding vectors |
| $\bar{X} = \bar{x}_1, \ldots, \bar{x}_{\frac{L}{r}}$ | embedding vectors of $X$ outputted by $f_{\text{enc}}(\cdot)$ |
| $q(\cdot)$ | quantization operator that replaces, e.g., $\bar{x}_1$ by its nearest neighbor in $\mathcal{E}$: $$g(\bar{x}_1) = \text{argmin}_{e_i \in \mathcal{E}} \|e_i - \bar{x}_1\|^2$$ |
| $g(\cdot)$ | maps prompt to a $d$-dimensional embedding vector |
| $f_{\text{dec}}(\cdot, \cdot)$ | decodes $L/r$ quantized embedding vectors in $\mathcal{E}$ back to text tokens, conditioning on prompt embedding generated by $g(\cdot)$ |

Table B.1: Mathematical notations used throughout the paper.

[3]https://github.com/meta-llama/llama-cookbook

# C   Details of Attention Weights Analysis

## C.1   Generated Responses

---

**Question 1**

**Prompt**: What is the positive difference between $120\%$ of 30 and $130\%$ of 20?

---

**Response of Our Latent Approach**:
`<boLatent><Latent-code-278> <Latent-code-416><eoLatent>` % of 30 and $130\%$ of 20. First, let's calculate $120\%$ of 30: $120\%$ of 30 = 1.2 x 30 = 36. Next, let's calculate $130\%$ of 20: $130\%$ of 20 = 1.3 x 20 = 26. Now, let's find the positive difference between 36 and 26: 36 - 26 = 10
The answer is 10.

---

**Response of Baseline (CoT)**: To find the positive difference between $120\%$ of 30 and $130\%$ of 20, we need to calculate each value first. $120\%$ of 30 is calculated as 1.2 * 30 = 36. $130\%$ of 20 is calculated as 1.3 * 20 = 26. Now, we need to find the positive difference between these two values. To do this, we subtract the smaller value from the larger value: 36 - 26 = 10.
The final answer is 10.

---

**Question 2**

**Prompt**: Mark has $50 in his bank account. He earns $10 per day at his work. If he wants to buy a bike that costs $300, how many days does Mark have to save his money?

---

**Response of Our Latent Approach**: `<boLatent> <Latent-code-123> <eoLatent>` need to calculate how much more money he needs to buy the bike and then divide that amount by his daily earnings. The bike costs $300, and Mark already has $50. So, he needs: $300 (bike cost) - $50 (current savings) = $250 (more money needed) Mark earns $10 per day. To find out how many days he needs to save, we divide the amount he needs by his daily earnings: $250 (more money needed) / $10 (daily earnings) = 25 days.
The answer is 25.

---

**Response of Baseline (CoT)**: To find out how many days Mark needs to save his money, we need to subtract the cost of the bike from the amount of money he has, then divide the result by the amount he earns per day. Mark has $50 and wants to buy a bike that costs $300. First, subtract the cost of the bike from the amount of money Mark has: $50 - $300 = -$250 Since Mark can't have a negative amount of money, we need to convert the negative amount to a positive one by adding the cost of the bike to the amount of money Mark has: $50 + $300 = $350 Now, divide the result by the amount Mark earns per day: $350 / $10 = 35
The final answer is 35.

---

# D  Other Text Generation Examples

---

**Example Generation of Our Latent Approach**

**Prompt**: Evaluate the expression $a^3 \cdot a^2$ if $a = 5$
**Generated Text**:
`<boLatent><Latent-code-130><eoLatent>` We can use the rule of exponents that states $a^m \cdot a^n = a^{m+n}$. So, $a^3 \cdot a^2 = a^{3+2} = a^5$. Now, we substitute $a = 5$ into the expression: $5^5$. $5^5 = 5 \cdot 5 \cdot 5 \cdot 5 \cdot 5 = 3125$. The answer is 3125.

---

**Example Generation of Our Latent Approach**

**Prompt**: Compute $\begin{pmatrix} 15 \\ 2 \end{pmatrix}$.
**Generated Text**: `<boLatent> <Latent-code-888> <Latent-code-744> <Latent-code-196> <Latent-code-92> <Latent-code-557> <eoLatent>` $\frac{15!}{2!(15-2)!} = \frac{15!}{2!13!}$ To simplify this expression, we can cancel out the common factors in the numerator and denominator: $\begin{pmatrix} 15 \\ 2 \end{pmatrix} = \frac{15 \times 14}{2} = 105$. The answer is 105.

---

**Example Generation of Our Latent Approach**

**Question**: How many vertical asymptotes does the graph of $y = \frac{2}{x^2+x-6}$ have?

**Answer**: `<boLatent><Latent-code-773> <Latent-code-514> <eoLatent>` the denominator, which is $x^2 + x - 6$. We can factor this quadratic expression as $(x + 3)(x - 2)$. Now, we can see that the denominator will be equal to zero when either $(x + 3) = 0$ or $(x - 2) = 0$. Solving for these values, we get $x = -3$ or $x = 2$. Therefore, the vertical asymptotes of the graph of $y = frac2x^2 + x - 6$ are $x = -3$ and $x = 2$. Since there are two distinct values of $x$ that make the denominator equal to zero, the graph has 2 vertical asymptotes. The answer is 2.

---

**Example Generation of Our Latent Approach**

**Question**: If there exists an ordered numbering of the nodes such that for each node there are no links going to a lower-numbered node, then there are no directed cycles in a directed graph. True or false?

**Answer**: `<boLatent><Latent-code-526> <eoLatent>` a directed graph. A directed cycle is a sequence of nodes in a graph, where each node is connected to the next node by a directed edge, and the last node is connected back to the first node by a directed edge. Now, let's consider the given condition: "If there exists an ordered numbering of the nodes such that for each node there are no links going to a lower-numbered node, then there are no directed cycles in a directed graph." This condition essentially means that if we number the nodes in a specific way, and there are no links going to a lower-numbered node, then there are no directed cycles in the graph. To see why this is true, let's consider what would happen if there were a directed cycle in the graph. If there were a directed cycle, then there would be a sequence of nodes where each node is connected to the next node by a directed edge, and the last node is connected back to the first node by a directed edge. However, if we number the nodes in the way described, and there are no links going to a lower-numbered node, then this directed cycle would be impossible. This is because the last node in the cycle would have to be connected to a lower-numbered node, which would violate the condition. Therefore, if we number the nodes in this way, then there can be no directed cycles in the graph. The answer is True.

# E Interpreability Examples

To examine the interpretability of the latent-codes, we use the decoder from the trained VQ-VAE to convert the latent-code back to the text space. In general, it is interpretable. We bolded the decoded part of the text from the latent code, which is delimited by the `<boLatent>` ... `<eoLatent>` tags. In general, we find the latent tokens capture long/verbose sentences which may trigger useful follow-up thinking pattern.

---

**Example Generation of Our Latent Approach**

**Prompt**: In a 8 fluid ounce bottle of Vitamin Water, there are 125 calories. How many calories would be contained in a 12 fluid ounce bottle? Express your answer in decimal form.

**Generated Text**:

`<boLatent>` **To convert the number from in in a 8 ounces** `<eoLatent>` we need to find the ratio of calories to fluid ounces in the 8 fluid ounce bottle and then multiply it by 12.

First, let's find the ratio of calories to fluid ounces in the 8 fluid ounce bottle:

Calories per fluid ounce = Total calories / Total fluid ounces

= 125 calories / 8 fluid ounces

= 15.625 calories per fluid ounce

Now, let's multiply this ratio by 12 fluid ounces to find the total calories in the 12 fluid ounce bottle:

Total calories in 12 fluid ounces = Calories per fluid ounce * Total fluid ounces

= 15.625 calories per fluid ounce * 12 fluid ounces

= 187.5 calories

The answer is 187.5.

---

Here, in the example above, the decoded latent token "To convert the number from in in a 8 ounces" captures the core concept of proportional conversion from the 8-ounce bottle to the 12-ounce bottle. Despite its grammar mistake, it still precisely states the mathematical operation needed to solve the problem.

---

**Example Generation of Our Latent Approach**

**Prompt**: What is the greatest common divisor of 315 and 108?

**Generated Text**:

`<boLatent>` **We find the greatest of CD of::63 common find** `<eoLatent>` we can use the Euclidean algorithm.

Here's how to do it:

1. Divide 315 by 108: $315 = 108 \times 2 + 99$

2. Divide 108 by 99: $108 = 99 \times 1 + 9$

3. Divide 99 by 9: $99 = 9 \times 11 + 0$

Since the remainder is 0, the GCD is the divisor in the last step, which is 9.

The answer is 9.

---

Here, in the example above, the decoded latent token "We find the greatest of CD of::63 common find" abstracts the initiation of the Euclidean algorithm, leading directly into the process that determines the GCD.

# F Additional Experiments

We present results of different approaches for fine-tuning a Llama-3.1-8B model on the DART-MATH (Tong et al., 2024) dataset. The observations are similar to those we presented in Section 4.2.

| Model (Dart-Math) | | In-Domain | | Out-of-Domain | | | | | Average |
|---|---|---|---|---|---|---|---|---|---|
| | | math | GSM8K | Fresh-Gaokao-Math-2023 | DeepMind-Mathematics | College-Math | Olympia-Math | TheoremQA | All Datasets |
| **Llama-3.1-8B** | Sol-only | 13.3 | 16.4 | 0.0 | 18.2 | 15.9 | 4.7 | 16.9 | 12.2 |
| | CoT | 43.1 | **84.5** | 30.7 | **47.8** | 45.7 | 10.1 | **21.2** | 40.4 |
| | iCoT | 35.2 | 61.8 | 30.0 | 30.6 | 37.6 | 8.3 | 19.5 | 31.8 |
| | Latent (Ours) | **43.2** (↑ +0.1) | 83.9 | **33.3** (↑ +2.6) | 44.7 | **47.1** (↑ +1.4) | **13.3** (↑ +3.2) | 20.3 | **40.8** (↑ +0.4) |

Table F.1: Our approach surpasses the iCoT and Sol-Only baseline when trained on the DART-MATH dataset (Tong et al., 2024), while marginally outperforming the CoT baseline.

| Model (Dart-Math) | | In-Domain (# of tokens) | | Out-of-Domain (# of tokens) | | | | | Average (# of tokens) |
|---|---|---|---|---|---|---|---|---|---|
| | | math | GSM8K | Fresh-Gaokao-Math-2023 | DeepMind-Mathematics | College-Math | Olympia-Math | TheoremQA | All Datasets |
| **Llama-3.1-8B** | Sol-only | 10.9 | 8.1 | 10.2 | 8.4 | 11.2 | 16.1 | 16.13 | 11.6 |
| | CoT | 522.7 | 181.0 | 628.8 | 343.2 | 486.3 | 893.7 | 648.3 | 529.1 |
| | iCoT | 397.1 | 118.6 | 440.8 | 227.9 | 321.9 | 614.4 | 485.7 | 372.3 |
| | Latent (Ours) | 489.1(↓ -6.4%) | 163.5 (↓ -9.7%) | 462.1(↓ -26.5%) | 265.6 (↓ -22.6%) | 396.3 (↓ -18.5%) | 801.3 (↓ -10.3%) | 591.3 | 452.7 (↓ -16%) |

Table F.2: The average number of tokens in the generated responses. Our approach generates shorter reasoning traces then the CoT baseline. ↓ -: Trace length reduction rate compared with CoT.

