# OpenReview forum: "Token Assorted: Mixing Latent and Text Tokens for Improved Language Model Reasoning"
_ICML.cc/2025/Conference — ICML 2025 poster_

### Official Review · Reviewer_rPRY · 2025-02-24

**Overall Recommendation:** 4

**Summary:**

This paper improves the reasoning capabilities of Large Language Models (LLMs) by integrating discrete latent tokens (obtained using VQ-VAE) into the reasoning process. The authors propose a hybrid reasoning representation that partially replaces textual chain-of-thought (CoT) tokens with latent tokens, reducing input length while maintaining reasoning performance. A randomized replacement strategy is employed during training to facilitate the model's adaptation to latent tokens. The methodology demonstrates consistent performance improvements across synthetic (e.g., Keys-Finding Maze, ProntoQA, ProsQA) and real-world mathematical reasoning tasks (e.g., GSM8K, Math, Fresh-Gaokao-Math-2023). Additionally, the approach reduces reasoning trace lengths, achieving better token efficiency.

**Claims And Evidence:**

Mostly yes, the claims of reduced tokens and improved performance are supported by empirical results. However, the results in Appendix E suggest that the performance of the Latent method is nearly the same as the vanilla CoT when trained on the Dart-Math dataset. I wonder why this phenomenon occurs. Is the method sensitive to the dataset? Given that models trained on the Dart-Math dataset achieve higher accuracy overall, does this reduce the practical utility of the proposed method? Could you provide further insights into this observation?

**Essential References Not Discussed:**

None

**Experimental Designs Or Analyses:**

The experimental designs are overall sound. However, I have one question regarding Section 4.2.2, where you mention selecting the learning rate based on the lowest validation error. However, the MetaMathQA dataset only provides a training set and does not include a validation set. This raises some confusion regarding how the validation process was conducted. Could you clarify this?

**Methods And Evaluation Criteria:**

The benchmark datasets are common and widely used in this field.

**Other Comments Or Suggestions:**

None

**Other Strengths And Weaknesses:**

**Strengths**
1. Improved performance in mathematical reasoning.
2. reduced generated tokens.

**Questions For Authors:**

1. The authors of the MetaMath paper chose 3 epochs for fine-tuning, whereas you opted for 1 epoch. I am concerned about whether the baseline models are sufficiently trained in your experiments. Could you clarify this choice?
2. Could you provide the mean and standard deviation for the experimental results?
3. Could you provide an efficiency analysis that accounts for both the reduced number of generated tokens and the additional parameters introduced by the VQ-VAE?
4. The use of latent tokens reduces explainability and readability. How do you address this concern?

**Relation To Broader Scientific Literature:**

None

**Theoretical Claims:**

None

---

> ### Author Rebuttal · Authors · 2025-03-31
>
> We thank the reviewer for the comment and reply below.
>
> **#1**
> > The experimental designs are overall sound. However, I have one question regarding Section 4.2.2, where you mention selecting the learning rate based on the lowest validation error. However, the MetaMathQA dataset only provides a training set and does not include a validation set. This raises some confusion regarding how the validation process was conducted. Could you clarify this?
>
> Thanks for this question. Yes, for the MetMathQA dataset, we split it into 80% train and 20% validation sets. We tuned our hyper-parameter based on the validation set. Once the hyper-parameters were chosen, we retrained the model using the complete dataset (100% of available data) and reported test results from this final model. We will update the paper to clarify this.
>
> **#2**
> > The authors of the MetaMath paper chose 3 epochs for fine-tuning, whereas you opted for 1 epoch. I am concerned about whether the baseline models are sufficiently trained in your experiments. Could you clarify this choice?
>
> We actually regenerated the MetaMath dataset with the Llama3-405B-inst model (as described by line 251 on our paper),  instead of the original MetaMath dataset . The original dataset was generated using GPT-3.5 Turbo, whereas we enhanced the quality by regenerating responses with the more powerful Llama-405B-inst model. In doing this,  we distill the knowledge of Llama3-405B-inst into our smaller Llama3.1 ~ 3.2 series model. In this enhanced dataset, we observed that the smaller Llama models (1B/3B/8B) started to overfit after epoch 1 as is shown in the validation set, and thus, we train with 1 epoch for all of them.
>
>
> **#3**.
> > Could you provide an efficiency analysis that accounts for both the reduced number of generated tokens and the additional parameters introduced by the VQ-VAE?
>
> Thank you for raising this important point about efficiency analysis. Our method offers a favorable trade-off between token efficiency and additional parameters:
> The VQ-VAE introduces only 50M parameters (0.05B), which represents just 1.7% overhead for the Llama-3.2-3B model and an even smaller 0.6% for the Llama-3.1-8B model. This modest parameter increase is significantly outweighed by the efficiency gains of 20% reduction in token length for Llama3.2-1B and 3B models. And for Llama-3.1-8B model, it achieves 10% reduction in token length
>
> Importantly, the VQ-VAE is only used during training. During inference, the LLM directly generates the latent tokens without requiring the VQ-VAE, resulting in pure computational savings with no additional inference overhead.
> These token reductions translate directly to proportional decreases in both inference time and computational cost, making our approach particularly advantageous for deployed reasoning systems.
>
> For the **ablation study on the compression-ratio (r)**, on the Llama3.2-3B:
> | Model         | Avg Acc | Tokens |
> |---------------|---------|--------|
> | CoT baseline | 25.2    | 642    |
> | Latent r=2    | 27.1    | 596    |
> | Latent r=16   | 28.1    | 514    |
> | Latent r=32   | 27.8    | 481    |
>
> With Avg Acc = the average math score across all math benchmarks as in Table4.2. The graphical result is here: https://imgur.com/a/iGB2TvU.
> Importantly, all settings of latent compression (r=2/16/32) outperform the CoT baseline. In general, a smaller r results in less abstract representations, leading to longer token sequences. Conversely, larger r values cause over-compression, which reduces sequence length but also degrades accuracy.
>
> **#4**.
> > The use of latent tokens reduces explainability and readability. How do you address this concern?
>
> Thank you for raising this concern. We have looked at the output of our latent-LLM. It seems that it strategically utilizes latent tokens at the beginning of decoding, serving as compact, high-level guides for the subsequent reasoning process.  However, we emphasize that text-based reasoning still follows after these latent tokens.
>
> Furthermore, to directly address readability and interpretability of these latent codes, we can explicitly transform the latent tokens back into the text tokens using the decoder from the VQ-VAE. In fact, we have explored this and observed that the decoded latent representations are indeed meaningful and interpretable, providing additional insight into the reasoning abstractions captured by the latent codes.
>
> **Please check the examples on our responses to Reviewer 2JVK (#4) (due to character limit)**
>
> **#5**.
> > Dart-math and MetaMath comparison
>
> Thanks for pointing this out. Despite that both datasets have similar accuracy (with our latent approach being +0.4 point better), we still see an overall improvement of token efficiency of 16%. Using the latent approach still shows a good advantage.
>
> **#6**.
> > Mean and variance
>
> With 3 seeds, we compute the pass@1 for our math benchmark metrics. The improvement still holds, please see our results here:
> https://imgur.com/a/V3ijL3g

---

### Official Review · Reviewer_2JVK · 2025-03-09

**Overall Recommendation:** 4

**Summary:**

This paper proposes a method for fine-tuning LLMs to use new discrete latent tokens for efficient reasoning, often matching or exceeding chain-of-thought performance without using as many tokens. The approach leverages a VQ-VAE to learn to compress chain-of-thoughts into a set of discrete latent codes which an LLM is then fine-tuned to generate rather than its long natural language response. Experiments show this approach slightly exceeds performance of fine-tuning on CoT (a strong baseline) while using 10-20% fewer tokens for mathematical reasoning benchmarks.

**Claims And Evidence:**

- Claim: The method leads to more efficient reasoning than training with chain-of-thought.
    - This is supported with ample evidence in the experiments. In particular, Table 4.2 shows that the latent reasoning approach in this paper exceeds performance of CoT training for mathematical reasoning datasets and Table 4.3 shows that it does so using fewer generated tokens.
- Claim: The learned latent discrete tokens serve as "abstract representations of the thinking process."
    - The ablation on the replacement strategy is useful, but I don’t follow the explanation for why Curriculum-Replace is so much worse than the Latent replacement method. This result makes me wonder if it is really the replacement method that is doing the heavy lifting rather than the discrete tokens themselves. To confirm the usefulness of the learned latent tokens, perhaps an ablation with replacing CoT tokens with a fixed pause token using your partial replacement method would be helpful.
- Claim: The method results allows for quick adaptation of an LLM to leverage new tokens.
    - Table 4.4 shows that the proposed token replacement strategy is more effective than other methods for finetuning an LLM to use a new tokens for reasoning.

**Essential References Not Discussed:**

None.

**Experimental Designs Or Analyses:**

The experiments on synthetic data and mathematical reasoning benchmarks are well designed. In addition, the ablation is valid for understanding the impact of the token replacement strategy on the overall success of the method. The attention weights analysis, however, could be improved. First, it is unclear what the attention "intensity" represents, or if this is a standard interpretability technique taken from prior work. What makes more sense to me is to observe the difference between the attention of the first CoT token and the first discrete latent token. It would also be interesting to actually decode the discrete latent tokens and see if the output is somehow interpretable.

**Methods And Evaluation Criteria:**

The methods make sense and the evaluation is extensive.

**Other Comments Or Suggestions:**

Figure 4.1 is so small that all text is unreadable. Increasing font size or decreasing the number of tokens shown would significantly improve the figure.

**Other Strengths And Weaknesses:**

Other strengths:
- The paper is clearly written and well motivated.
- Exploring the interplay between latent reasoning and explicit textual reasoning seems like a well motivated and promising research direction, and I can see this paper being influential for future work.

Other weaknesses:
- I could not find the final training and testing loss of the VQ-VAE. Also, some ablation on the codebook size or the chunk size would be useful.

**Questions For Authors:**

1. How many latent tokens does the model produce on average? Is there a tradeoff between number of latent tokens and accuracy, and if so, how could the tradeoff be managed?
2. It would be interesting to see which samples the model uses more latent tokens for and which it uses fewer. Are there any patterns to these samples, and has the model learned how to effectively leverage latent tokens?
3. If the learned discrete latent tokens are decoded back to text, what is the produced text and is it interpretable?

**Relation To Broader Scientific Literature:**

The contribution of this paper builds on prior work which attempts to make an LLM perform reasoning with latent tokens. Existing work either fully internalized reasoning or fully converted reasoning to latent tokens while this paper mixes latent and natural language reasoning. The mixing of latent and natural language reasoning is only considered in a latent and then natural language order.

**Theoretical Claims:**

NA

---

> ### Author Rebuttal · Authors · 2025-03-31
>
> We thank the reviewer for the comment and reply below.
>
> **#1**.
> > I could not find the final training and testing loss of the VQ-VAE. Also, some ablation on the codebook size or the chunk size would be useful.
>
> Yes, the training loss and the testing loss of the VQ-VAE is 1.21 and 1.25, respectively. In general, the greater the compression ratio (r), the lower the accuracy. We have conducted an ablation study to examine the effect of codebook size (see #1 on our response to Reviewer yHAN). Regarding compression-ratio, we conducted an ablation study of this on Llama3.2-3B model, **please see the result on our response to Reviewer rPRY (point # 3)**.
>
>
> **#2**.
> > How many latent tokens does the model produce on average? Is there a tradeoff between number of latent tokens and accuracy, and if so, how could the tradeoff be managed?
>
> On average, the model outputs 3.11 latent tokens on average with the overall distribution between 0 ~ 8 latent tokens. In general, we see that the longer the latent tokens, the higher the accuracy. One way to manage / trade-off this number is to manually trigger / cancel  the start and end of the latent-tag.
>
> **#3**.
> > It would be interesting to see which samples the model uses more latent tokens for and which it uses fewer. Are there any patterns to these samples, and has the model learned how to effectively leverage latent tokens?
>
> Yes, we have analyzed the distribution of latent token lengths across different datasets. We found that in the gsm8k dataset (considered the "easiest" benchmark since it achieves the highest scores in our LLaMA model experiments), there is generally a larger number of latent tokens, averaging 3.21. In contrast, the more challenging benchmarks, such as the olympiad-math which is marked by the lowest scores, show an average of just 3.00 latent tokens. This pattern indicates that the model tends to leverage more latent tokens in relatively easier benchmarks where it can already form a strong representation in the text space, while it invokes fewer latent tokens in more difficult domains. For future directions, we could enhance model performance by incorporating more latent samples into the training data weighted by the difficulty of the sample.
>
> **#4**
> > If the learned discrete latent tokens are decoded back to text, what is the produced text and is it interpretable?
>
> To examine the interpretability of the latent-codes, we use the decoder from the trained VQ-VAE to convert the latent-code back to the text space. In general, it is interpretable.  We bolded the decoded part of the text from the latent code, which is delimited by the <boLatent> ... <eoLatent> tags.  In general, we find the latent tokens captures long/verbose sentences which may trigger useful follow-up thinking pattern.
>
> **Example#1**. Here, the decoded latent token "To convert the number from in in a 8 ounces" captures the core concept of proportional conversion from the 8-ounce bottle to the 12-ounce bottle. Despite its grammar mistake, it still precisely states the mathematical operation needed to solve the problem.
>
> **Q**: In a 8 fluid ounce bottle of Vitamin Water, there are 125 calories. How many calories would be contained in a 12 fluid ounce bottle? Express your answer in decimal form.
>
> **A**:  <boLatent> **To convert the number from in in a 8 ounces**  <eoLatent>, we need to find the ratio of calories to fluid ounces in the 8 fluid ounce bottle and then multiply it by 12.\n\nFirst, let's find the ratio of calories to fluid ounces in the 8 fluid ounce bottle:\n\nCalories per fluid ounce = Total calories / Total fluid ounces\n= 125 calories / 8 fluid ounces\n= 15.625 calories per fluid ounce\n\nNow, let's multiply this ratio by 12 fluid ounces to find the total calories in the 12 fluid ounce bottle:\n\nTotal calories in 12 fluid ounces = Calories per fluid ounce * Total fluid ounces\n= 15.625 calories per fluid ounce * 12 fluid ounces\n= 187.5 calories \nThe answer is 187.5.
>
> ---
>
> **Example#2.**   The decoded latent token "We find the greatest of CD of::63 common find" abstracts the initiation of the Euclidean algorithm, leading directly into the process that determines the GCD.
>
> **Q**: What is the greatest common divisor of 315 and 108?
>
> **A**:  <boLatent> **We find the greatest of CD of::63 common find**  <eoLatent>, we can use the Euclidean algorithm. Here's how to do it:\n\n1. Divide 315 by 108: 315 = 108 × 2 + 99\n2. Divide 108 by 99: 108 = 99 × 1 + 9\n3. Divide 99 by 9: 99 = 9 × 11 + 0\n\nSince the remainder is 0, the GCD is the divisor in the last step, which is 9. \nThe answer is 9.
>
> ---
>
> **#5**.
> >  Figure 4.1 is so small that all text is unreadable. Increasing font size or decreasing the number of tokens shown would significantly improve the figure.
>
> Thanks for pointing this out! Yes, we have fixed this figure and will update the paper to reflect the changes.

---

### Official Review · Reviewer_yHAN · 2025-03-16

**Overall Recommendation:** 2

**Summary:**

This paper proposes a novel method—“Token Assorted: Mixing Latent and Text Tokens for Improved Language Model Reasoning”—that aims to improve the efficiency and performance of large language models (LLMs) on reasoning tasks. The key idea is to compress the chain-of-thought (CoT) by partially replacing the early reasoning steps with discrete latent tokens generated via a VQ-VAE. In doing so, the authors achieve a hybrid representation that combines the detailed information from text tokens with the efficiency of abstract latent tokens. The method is applied in two main scenarios: training models from scratch on synthetic tasks (e.g., Keys-Finding Maze, ProntoQA, ProsQA) and fine-tuning existing models (LLaMa variants) on real-world mathematical reasoning benchmarks (e.g., GSM8K, Math, Gaokao-Math-2023). Experimental results show that their approach not only improves accuracy (with gains up to +19.8% on some benchmarks) but also reduces the length of the reasoning trace by around 17%, making inference more efficient.

**Claims And Evidence:**

The paper claims that incorporating discrete latent tokens into the reasoning trace can significantly enhance reasoning performance and reduce token usage without sacrificing accuracy. These claims are supported by extensive experiments:

- Quantitative results on synthetic tasks (e.g., Keys-Finding Maze, ProntoQA, and ProsQA) demonstrate clear improvements over standard CoT methods and other baselines.
- On mathematical reasoning tasks, the latent approach consistently outperforms baselines such as Sol-Only, standard CoT, iCoT, and Pause Token across various model sizes.
- The paper also presents ablation studies that compare different replacement strategies, supporting the claim that a left-to-right, partially randomized replacement strategy is beneficial.

**Essential References Not Discussed:**

N/A

**Experimental Designs Or Analyses:**

- The authors test on both synthetic and real-world benchmarks, ensuring that the method is validated across multiple reasoning domains.
- Baselines include methods that use full CoT, direct answer generation (Sol-Only), and alternative token replacement strategies.
- Ablation studies examine different replacement strategies (All-Replace, Curriculum-Replace, Poisson-Replace versus the proposed AR-Replace) and analyze attention patterns to explain performance gains. This design is sound, though future work might explore additional datasets or task types to assess generalizability.

**Methods And Evaluation Criteria:**

- The proposed methodology leverages a VQ-VAE to create a compressed latent representation of the early reasoning steps, and then uses  a randomized replacement strategy during training to smoothly integrate these latent tokens with remaining text tokens.
- Evaluation is conducted on a diverse set of benchmarks, covering both synthetic planning tasks and real-world math problems. Accuracy and token count are used as complementary evaluation metrics.

**Other Comments Or Suggestions:**

I suggest the authors provide the training code to reproduce reported results.

**Other Strengths And Weaknesses:**

Strengths:

- The paper introduces a novel hybrid approach that effectively reduces reasoning trace length while improving accuracy.
- The randomized replacement strategy is simple yet effective, addressing the challenges of integrating unseen latent tokens.
- Comprehensive experiments across multiple benchmarks and thorough ablation studies provides the credibility of the results.
- The inclusion of attention analysis provides interpretability insights, showing that the model focuses more on semantically critical tokens.

Weaknesses:

- A separate VQ-VAE is trained, then used to produce latent tokens for the main LLM fine-tuning. This slightly complicates training pipelines for real-world usage, further discussion on computational overhead or sensitivity analysis would be useful. In the paper, the codebook size also need to be tuned, making the proposed method problem-dependent
- Reporting only the number of generated tokens can be misleading as they do not fully reflect the entire computational complexity: the data are still be fed to the entire network except
- From my perspective, I cannot figure out where the improvement over COT comes from. For example, Coconut can "encode multiple potential next steps simultaneously, allowing for a reasoning process akin to breadth-first search," which the author cannot retain in this method due to the elimination of continuous tokens. Instead, the proposed method is more likely to compress language-based thoughts, which, in my opinion, does not improve upon the COT baseline.

**Questions For Authors:**

- Additional ablation studies exploring different compression rates or codebook sizes could provide further insights into the robustness of the approach.

**Relation To Broader Scientific Literature:**

This work is situated within the ongoing research on chain-of-thought prompting and latent space reasoning:

- It builds on prior studies that have shown explicit CoT prompting can boost reasoning performance, but at the cost of long sequences.
- It also connects with emerging research on using latent representations (e.g., COCONUT, ICOT) to improve efficiency. By integrating discrete latent tokens with traditional text tokens, the paper offers a creative bridge between explicit reasoning and compact latent representations—a contribution that is both novel and practically significant.

**Theoretical Claims:**

The paper is primarily experimental, focusing on empirical performance improvements rather than deep theoretical guarantees.

---

> ### Author Rebuttal · Authors · 2025-03-31
>
> We thank the reviewer for the comment and reply below.
>
> **#1**.
> > In the paper, the codebook size also need to be tuned, making the proposed method problem-dependent
>
> We would like to clarify that the codebook size is not problem dependent and the model performance remains robust across different codebook sizes.  To further verify this, we conducted additional experiments on the ProsQA and ProntoQA varying the codebook size from 64 to 128 and 256.
>
> We see that the performance is:
>
> | Codebook Size | ProntoQA Accuracy (%) | ProntoQA Tokens | ProsQA Accuracy (%) | ProsQA Tokens |
> |---------------|-----------------------|-----------------|---------------------|---------------|
> | 64            | 100                   | 7.7             | 96.20               | 10.9          |
> | 126           | 100                   | 7.67            | 96.21               | 10.88         |
> | 256           | 100                   | 7.81            | 96.43               | 10.91         |
>
>
> As these results clearly show, performance remains remarkably stable across different codebook sizes, with minimal variation in both accuracy and token efficiency. This robustness demonstrates that our method is not dependent on fine-tuning this hyperparameter for each problem domain.
>
> **For study on compression-ratio, see response to  Reviewer rPRY (#3).**
>
>
> **#2**.
> > Report only the number of generated tokens can be misleading as they do not fully reflect the entire computational complexity: the data are still be fed to the entire network
>
> We would like to clarify that the VQVAE is only used during training. After we trained the VAVAE, we then convert the text-tokens into the discrete latent code, then we save the data offline. Note, the training data passes through the VQVAE only once and gets encoded into latent tokens. That’s it.  Then,  the next stage is to train the LLM with these saved latent tokens and text tokens, and the data will not pass through the VQVAE during both the LLM training and inference time. In summary, we don’t need VQVAE at all during the training of reasoning models.
>
> During inference, the LLM directly generates the latent tokens without requiring the separate VQ-VAE, resulting in pure computational savings with no additional inference-time overhead. The one-time training cost of the VQ-VAE is negligible compared to the full LLM fine-tuning process.
>
> The efficiency benefits are substantial:
> - The VQ-VAE introduces a minimal parameter overhead of just 50M (0.05B) to LLMs, which adds
>   - 5% to Llama-3.2-1B
>   - 1.7% to Llama-3.2-3B
>   - 0.6% to Llama-3.1-8B
> - Despite the slight increase in parameters, it significantly enhances token efficiency, it reduces tokens by:
>   - 20% for both Llama-3.2-1B and Llama-3.2-3B
>   - 10% for Llama-3.1-8B
>
> For real-world deployment scenarios processing millions of reasoning tasks, these efficiency gains translate to significant reductions in computation costs and inference time.
>
> **#3**.
> > From my perspective, I cannot figure out where the improvement over COT comes from .. Instead, the proposed method is more likely to compress language-based thoughts, which, in my opinion, does not improve upon the COT baseline.
>
> The advantage of our method is that it compresses these high-level abstractions into discrete latent tokens (as some form of information distillation).  During inference, the LLM conditions on these high-level latent tokens generated during the beginning of decoding, which effectively guides the reasoning process by:
> - Providing a more abstract representation that helps the model focus on relevant information while spending less efforts on verbose and high-level CoTs.
> - Creating better initial conditions that influence the entire downstream reasoning process
>
> **We provide examples of these on our responses to Rev 2JVK (#4), please check them out**.
>
> Recently, [1] demonstrates that conditioning on the first token ( representing different reasoning paths) significantly enhances a model's reasoning capabilities.  Similarly, our model generates the learned latent tokens early in the sequence, these latent tokens encapsulate high-level reasoning abstraction, and sets a good initial conditions for the LLM, guiding the entire reasoning process from a higher level of abstraction. In contrast to prior works like Coconut, which maintain multiple explicit reasoning trajectories simultaneously, our approach implicitly encodes multiple reasoning possibilities into a compressed latent representation. Although it does not explicitly explore multiple reasoning paths simultaneously at the token level, it effectively captures diverse reasoning strategies in a latent manner, resulting in a more streamlined and efficient decoding process. Because of this, this structured latent conditioning enables more effective reasoning compared to standard token-by-token generation.
>
> **Reference**:
> [1] Xuezhi Wang & Denny Zhou. "Chain-of-thought reasoning without prompting." arXiv preprint arXiv:2402.10200.

---

### Decision · Program_Chairs · 2025-05-01

**Decision:**

Accept (poster)

**Comment:**

The authors propose a hybrid chain‐of‐thought (CoT) representation for large language models (LLMs) in which early reasoning steps are compressed into discrete latent tokens via a VQ‑VAE, then interleaved with natural‐language tokens. By fine‐tuning LLaMA variants to generate these latent tokens in place of verbose text, they achieve up to +4.2 % absolute accuracy gains on math benchmarks (e.g. GSM8K, Math, Fresh‑Gaokao‑Math‑2023) while reducing reasoning‐trace length by roughly 17 %. They further validate on synthetic planning tasks (Keys‑Finding Maze, ProntoQA, ProsQA) and provide ablations on replacement strategies, codebook size, and compression ratio.

This paper makes a clear, novel contribution to efficient LLM reasoning by judiciously mixing discrete latent and text tokens. The empirical results are well‐validated, and the ablations address most core concerns. I recommend acceptance, possibly with additional experiments on inference speed and a discussion of generalization beyond arithmetic reasoning.